# Role of Cyclins and Cytoskeletal Proteins in Endometriosis: Insights into Pathophysiology

**DOI:** 10.3390/cancers16040836

**Published:** 2024-02-19

**Authors:** Marcin Szymański, Klaudia Bonowicz, Paulina Antosik, Dominika Jerka, Mariola Głowacka, Małgorzata Soroka, Kerstin Steinbrink, Konrad Kleszczyński, Maciej Gagat

**Affiliations:** 1Department of Histology and Embryology, Collegium Medicum in Bydgoszcz, Nicolaus Copernicus University in Torun, 85-092 Bydgoszcz, Poland; md.marcinszymanski@gmail.com (M.S.); klaudia.bonowicz@cm.umk.pl (K.B.); dominika.jerka@cm.umk.pl (D.J.); 2Faculty of Medicine, Collegium Medicum, Mazovian Academy in Płock, 08-110 Płock, Poland; m.glowacka@mazowiecka.edu.pl (M.G.); m.soroka@mazowiecka.edu.pl (M.S.); 3Department of Clinical Pathomorphology, Collegium Medicum in Bydgoszcz, Nicolaus Copernicus University in Torun, 85-094 Bydgoszcz, Poland; paulina.antosik@cm.umk.pl; 4Department of Dermatology, University of Münster, Von-Esmarch-Str. 58, 48149 Münster, Germany; kerstin.steinbrink@ukmuenster.de (K.S.); konrad.kleszczynski@ukmuenster.de (K.K.)

**Keywords:** endometriosis, cyclins, cytoskeletal proteins, EMT, TGF-β

## Abstract

**Simple Summary:**

This article explores the cellular and molecular intricacies of endometriosis, a gynecological condition marked by the abnormal growth of endometrium-like tissue outside the uterus. Emphasizing the roles of cyclins and cytoskeletal proteins, particularly in Epithelial–Mesenchymal Transition (EMT), the review highlights their diverse functions in cell cycle control, proliferation, evasion of apoptosis, and angiogenesis. The analysis underscores the interconnected pathways through which these proteins converge, influencing the genesis and progression of endometriosis. Understanding these complexities reveals insights into the disease’s causes and offers promise for targeted therapeutic approaches, ushering in a new era in managing this challenging disorder.

**Abstract:**

Endometriosis is a gynecological condition where endometrium-like tissue grows outside the uterus, posing challenges in understanding and treatment. This article delves into the deep cellular and molecular processes underlying endometriosis, with a focus on the crucial roles played by cyclins and cytoskeletal proteins in its pathogenesis, particularly in the context of Epithelial–Mesenchymal Transition (EMT). The investigation begins by examining the activities of cyclins, elucidating their diverse biological roles such as cell cycle control, proliferation, evasion of apoptosis, and angiogenesis among ectopic endometrial cells. A comprehensive analysis of cytoskeletal proteins follows, emphasizing their fundamental biological roles and their specific significance to endometriotic cell features. This review sheds light on the interconnected pathways through which cyclins and cytoskeletal proteins converge, contributing to the genesis and progression of endometriosis. Understanding these molecular complexities not only provides insight into the underlying causes of the disease but also holds promise for the development of specific therapeutic approaches, ushering in a new era in the management of this devastating disorder.

## 1. Introduction

Endometriosis is the leading cause of infertility in women of reproductive age [1]. It involves the presence of ectopic endometrium outside the uterine mucosa, mainly in the peritoneal cavity [2], extending to the ovaries, fallopian tubes, connective tissues, and mucous membranes in the pelvic regions [3]. Symptoms include heavy menstrual bleeding, gastrointestinal symptoms, fatigue, painful ovulation, irregular menstrual cycles, urinary symptoms, and, notably, chronic pelvic pain due to adhesions and fibrosis [4,5,6]. For clarity and precision, in this discussion, the term ‘endometrium’, representing the Latin and common name of this tissue structure, will be used when referring to the tissue in its normal anatomical position (eutopic). Nevertheless, ‘endometrium-like tissue’ will describe a tissue structure resembling the endometrium but located outside its typical anatomical position. The terms ectopic endometrium and endometrium-like tissue, as they are unambiguous, will be used interchangeably.

The etiology of endometriosis is a complex and multifaceted medical phenomenon. This condition may result from anatomical distortions due to adhesions or fibrosis, as well as endocrine or immunological mechanisms, which can sometimes overlap. In some cases, the various pathophysiological mechanisms in endometriosis seem to operate through poorly understood pathways [1]. The mechanism known as Epithelial–Mesenchymal Transition (EMT) plays a crucial role in explaining the genesis of endometriosis. This process significantly influences the development of scar tissue and the transformation of epithelial cells into mesodermal cells. One important player in this intricate process is TGF-β1 (Transforming Growth Factor Beta 1). According to the theory behind the EMT process, endometrial cells undergo changes facilitated by TGF-β1, giving them the capability to spread and infiltrate other bodily tissues. Altered endometrial cells can migrate to the abdominal cavity or other regions outside the uterine mucosa, leading to the formation of endometrial lesions. This concept enhances our comprehension of the intricate mechanisms that underlie the onset of this challenging condition [7,8]. It is crucial to emphasize that discussions surrounding the mechanisms of endometriosis remain controversial, with various theories continually under examination and analysis in light of new scientific discoveries. The subject of considerable deliberation in the field of endometriosis pathophysiology includes reports on clonality and the emerging Genetic–Epigenetic (GE) theory. The inclusion of the GE theory adds an additional layer of complexity to our understanding of this condition. Ongoing scientific exploration and the scrutiny of diverse theories, including the Genetic–Epigenetic perspective, contribute to a more comprehensive understanding of the intricate factors influencing the development and manifestation of endometriosis [9].

To effectively treat and assess the risks associated with endometriosis, a disease with a complex physiological basis, it is necessary to have a comprehensive understanding of the mechanisms of its formation and development [10]. This entails understanding how cells from the normal uterine lining navigate to abnormal sites, undergo excessive multiplication, evade immune and apoptotic processes, and acquire the necessary blood supply and essential nutrients, ultimately leading to the growth of abnormal tissue. In the most severe cases, this tissue can infiltrate neighboring areas or even take on an endocrine function [11]. Therefore, it is essential to identify and describe the molecules that play a role in regulating the cascades of mechanisms contributing to the acquisition of traits such as increased migration, proliferation, motility, and invasion in these normal cells. These factors are emphasized as being crucial in the context of the development of this significantly life-impairing disease [12,13].

The aim of this review is to highlight the significance of cyclins and cytoskeletal proteins in the context of the mentioned pathophysiological processes. The review focuses on the properties of cellular processes and their impact on the entire ectopic tissue during the course of endometriosis.

## 2. Materials and Methods

### 2.1. Literature Search Strategy

In the pursuit of a comprehensive understanding of the molecular mechanisms and cellular processes underlying endometriosis, we conducted an exhaustive systematic literature search. The literature was selected in order to establish causal relationships between variables. In the studies that were used, attention was paid to the investigators who were seeking to understand the functional roles of cytoskeletal proteins and cyclins and their impact on cancer (endometriosis if available) development and progression. The primary focus was on recent publications available via PubMed. The search was refined using specific keywords, including “cyclins”, “cytoskeletal proteins”, “cytoskeleton”, “endometriosis”, and “EMT”. We deliberately concentrated on studies published within the past 5 years.

### 2.2. Bibliography Assembly

The bibliography assembled for this study includes both the latest research and older references. Recommendations from recent articles guided the inclusion of studies spanning from the earliest publication in 1991 (Glotzer et al.) to the most recent research in 2023 (Zhao et al.). This intentionally broad timeframe allows for the inclusion of diverse research aspects from over the years, ensuring the consideration of seminal statements and foundational research.

### 2.3. Screening Process and Data Extraction

The screening process employed a two-tiered approach, initially scrutinizing titles and abstracts for relevance to the molecular intricacies of endometriosis. Subsequently, selected articles underwent a thorough full-text assessment. Data extraction involved the careful documentation of study design, key findings, and methodologies employed. This rigorous and systematic approach ensures the synthesis of information.

### 2.4. Inclusion of Studies and Exclusion Criteria

The articles included in this investigation were based on a comprehensive methodology aimed at elucidating the roles of cytoskeletal proteins and cyclins in the context of endometriosis and cancer. Experimental approaches, including in vitro cell culture studies, were employed to investigate how variations in these proteins influenced cellular behaviors. Molecular analyses, such as Western blotting and immunofluorescence microscopy, were utilized to assess both the quantitative and spatial aspects of protein expression and interactions. Additionally, the studies delved into the intricate signaling pathways associated with cytoskeletal dynamics and the Epithelial–Mesenchymal Transition (EMT), highlighting a meticulous examination of the underlying molecular mechanisms.

To ensure the robustness and relevance of the collected data, certain studies were excluded based on inappropriate study design, limited study availability, and irrelevance to the research topic. Specifically, publications were excluded if they lacked detailed information on the experimental protocol, including cell culture conditions and appropriate experimental controls. The included articles not only spanned a wide chronological spectrum but also encompassed a variety of primary statements and studies that had laid the groundwork for subsequent research endeavors.

### 2.5. Flowchart Representation

The numerical representation of the position selection process with exclusion criteria is depicted in the following flowchart (Figure 1).

## 3. Primary Biological Functions and Regulatory Mechanisms of Cyclins in Cell Cycle Control

Cyclins constitute a family of proteins pivotal in regulating the cell cycle, exerting direct control over proliferation, influencing cell differentiation, function, and cell death. Notably, their concentration in the cell undergoes cyclic variations rather than remaining constant [14,15]. In cytophysiology, they are best characterized as proteins binding to Cyclin-Dependent Kinases (CDKs), a group of enzymes activated by cyclins. Upon activation, CDK–cyclin complexes phosphorylate specific target proteins, enabling cells to progress through distinct stages of the cell cycle [16]. Cells possess specialized cytophysiological regulatory mechanisms governing cyclin levels. In particular, CDKs facilitate the degradation of cyclins through proteasomal pathways, initiated via ubiquitination [17]. Cyclins can be categorized based on the specific phase of the cell cycle in which they predominantly exert their essential functions. In the context of somatic cell biology, it is crucial to recognize the four distinct phases: G1 (Gap 1), S phase (Synthesis), G2 (Gap 2) phase (collectively known as interphase), and M phase (Mitosis) (Figure 2) [18,19,20].

Furthermore, CDK activity is tightly controlled by two primary families of Cyclin-Dependent Kinase Inhibitors (CDKIs): the CIP/KIP and the INK4 family [21]. The first directly interacts with both cyclin and CDK subunits, modulating the activity of CDK–cyclin complexes. On the other hand, INK4 family members specifically target CDK4 or CDK6, inhibiting their kinase activity by disrupting their interaction with type D cyclins [22]. In general, CDKIs curtail cell proliferation by neutralizing the catalytic activity of CDKs. Excessive cyclin production or a deficiency of CDKI expression disrupts the cellular system, leading to cell cycle progression and the deregulation of the whole cycle [23].

Comprehending cyclins involvement in endometriosis highlights their crucial role in regulating cellular proliferation. Advances in CDKIs target aberrant cell proliferation in cancer. Recent progress in combination treatments addresses initial limitations, showing clinical promise. The FDA-approved third CDKI generation, with selective CDK4/6 inhibitors, exhibits efficacy in treating diverse cancers, either standalone or in combination therapies [23,24]. The INK4 family of proteins plays a crucial role in G1 arrest by facilitating the redistribution of CIP/KIP proteins and impeding the activity of the cyclin E-CDK2 complex. CDK4/6 activity is tightly regulated by direct interaction with INK4 family members, including p16INK4A, p15INK4B, p18INK4C, and p19INK4D [25]. Upon binding to CDK4/6, INK4 family members induce a conformational change that disrupts the cyclin binding site, effectively suppressing the formation of the active cyclin D–CDK4 complex. which plays a key role in the molecular decision-making that determines cell division (Table 1) [26].

During the G1 phase, pivotal factors include external elements, such as interleukins like IL-2, which stimulate cell division, and IL-3, encouraging the differentiation of multipotent cells. Additionally, interferons including INF-γ act as significant modulators, inhibiting proliferation within this phase [23,27,28]. TGF-β, often categorized as a cytokine, exerts a wide-ranging regulatory influence by indirectly affecting various other proteins. It can have dual effects, reducing the incidence of cancer in some contexts, while in others, it may promote cancer progression. TGF-β’s intricate and context-dependent actions contribute to its complex role in cellular processes, including those related to cancer [29]. In its capacity as a tumor suppressor, TGF-β mediates antiproliferative effects across a broad spectrum of cell types [30]. CDK inhibitors have paved the way for the development of targeted drugs, including palbociclib, ribociclib, and abemaciclib, classified as CDK4/6 inhibitors. These drugs effectively arrest cancer cells in the G1 phase (Figure 2) [31]. However, it is crucial to acknowledge that during development, the levels, expression patterns, and interactions of these cyclins hold significant embryological importance. These factors play a fundamental role in essential processes, such as gastrulation and neurulation [32]. G1/S cyclins form a functionally interconnected group with G1 cyclins. These cyclins are essential factors enabling the cell’s transition from the G1 phase to the S phase of the cell cycle, exerting a substantial impact on the final phases of this transition. Within the G1/S cyclin group, E cyclins, specifically E1 and E2, are notable members that predominantly form complexes with CDK2 proteins (Table 1) [33].

Furthermore, the S cyclins are remarkably active during the synthesis phase. This group encompasses cyclin A, which is responsible for leading the process of DNA replication. At the outset of this cycle phase, cyclin A forms complexes with CDK2. As the phase progresses, it also has the capability to bind to CDK1, signifying its role in facilitating the transition of the cell towards the end of this phase (Figure 2). This dual binding capacity underscores its significance in governing the transition of a somatic cell into the division phase (M phase) [34,35,36]. In the S phase, internal factors, such as p21, TGF-β, and Emi1, play crucial roles. Emi1 serves as an inhibitor of the Anaphase-Promoting Complex/Cyclosome (APC/C). This phase is significant in the context of the actions of nucleoside analogs like gemcitabine and cytarabine, which impede DNA replication [37,38,39].

Within the G2 phase, a myriad of intracellular factors orchestrate critical regulatory functions. CDC25, a phosphatase, plays a central role by activating CDK1 through dephosphorylation, thereby facilitating cell cycle progression. Additionally, kinases such as Wee1 and Myt1 exert inhibitory control by phosphorylating CDK1. Furthermore, the 14-3-3σ protein assumes a pivotal role by sequestering cyclin B1-CDK1 complexes in the cytoplasm, effectively inhibiting the initiation of mitosis [40]. In this stage, adavosertib serves as a Wee1 inhibitor, impacting cell cycle regulation at the G2/M checkpoint by targeting this kinase (Figure 2) [41].

In the intricacies of mitosis, cyclin B, through its interaction with CDK1, crucially regulates the M phase, leading to mitotic events [42]. Cyclin B degradation at mitosis completion is a vital step, marking an exit from mitosis and enabling the transition to the next cell cycle phase. This underscores its pivotal role, serving as a central component in directing mitotic events [43] (Table 1).

It is pertinent to note that some cyclins, exemplified by H, C, and T, participate in essential processes such as transcription and DNA repair. Notwithstanding their specific functionalities, these cyclins do not fall directly within the classification of cell cycle determinants [44]. Specifically, cyclin C is often labeled as “non-cyclic” due to its involvement in cellular processes extending beyond typical cell cycle regulation [45]. Their inclusion in the cyclin family is primarily attributed to their capability to bind with CDKs [46].

Hence, the functions of some cyclins remain unclear, and there are insufficient data to fully comprehend their synthesis and roles. This knowledge gap can lead to uncertainties in research interpretation, especially due to variations in traditional naming conventions for certain cyclins [47,48,49,50,51].

**Table 1 cancers-16-00836-t001:** Cyclin-Dependent Kinase (CDK) networks: regulatory partners and functional roles.

Cyclin	Dependent CDK	Other Binding Proteins	Function
A	CDK1, CDK2	CDC6	S-G2 Transition, G2 Phase Progression. Pagano et al. [52]
B	CDK1	TGFBR2	Entry and Exit of M Phase. Hayles et al. [53]
C	CDK8	MED Proteins	Transcription Initiation. Rickert et al. [54]
D	CDK4, CDK6	p21	pRb Phosphorylation, G1 Progression. Matsushime et al. [55]
E	CDK2	p21, p27	pRb Phosphorylation, G1-S Phase Transition. Koff et al. [56]

**Figure 2 cancers-16-00836-f002:**
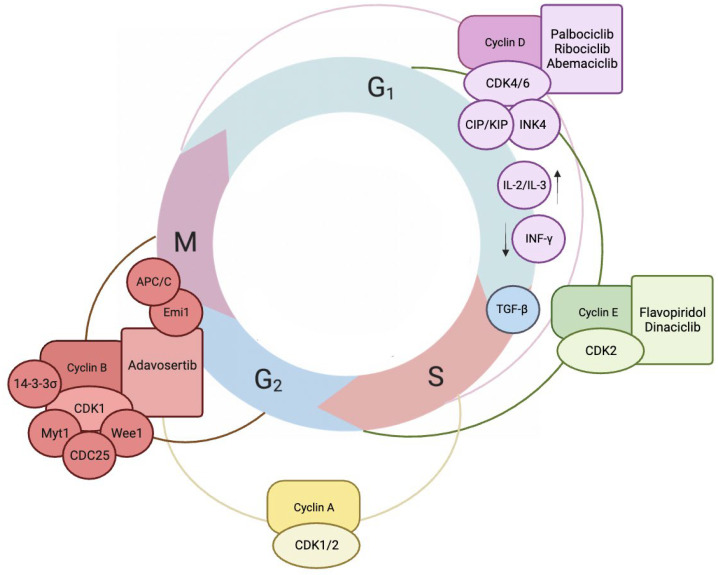
The integrated illustration showcases the intricate interplay and regulatory connections between various proteins, emphasizing the dynamic interactions between cyclins, CDKs, and CDKIs throughout the cell cycle [21,49,57,58] (designed with BioRender: https://biorender.com/ (accessed on 8 November 2023)).

### Deciphering the Pivotal Role of Cell Cycle Proteins in Endometriosis Pathogenesis

Current medical knowledge reveals elevated expression levels of specific cyclins within ectopic endometrium-like tissues in the context of endometriosis, contrasting with their expression in normal endometrium or healthy adjacent tissues [59]. These disparities primarily manifest as elevated tissue-level expression of not only particular cyclins but also cyclin-CDK complexes. This observation may imply the presence of distinct mechanisms contributing to cell cycle dysregulation in the context of endometriosis [60]. Cyclins and cyclin–CDK complexes are associated with increased cell proliferation in endometriosis and reduced susceptibility to apoptosis. In essence, lesions of endometrial cells exhibit augmented levels of cyclins, including stimulatory ones like cyclins A2, B1, B2, and D1, coupled with diminished expression of cell cycle inhibitors, such as p21, p27, p16, p18, and p19. These factors collectively contribute to increased cell division and reduced inhibition [61,62]. Recent studies suggest a role for cyclins in the Epithelial–Mesenchymal Transition (EMT). A study by Wang et al. indicated that the reduction in Cyclin A correlated with heightened cytoskeletal reorganization and enhanced cell migration in normal mammary epithelial cells [63].

In the mentioned cyclin triad, cyclin B1’s functions are well recognized. Certain studies indicate that cyclin A2 has a partial impact on the proliferative potential of ectopic endometrium, depending on the patient’s physiological condition. Its expression intensifies in stromal cells, particularly under the influence of sex hormones [64]. Additionally, its overexpression has been reported in vitro in endometrial cancer [65]. However, evidence regarding the role of cyclin B1 is more substantial. A study by Tang et al. of tissues from endometriosis patients demonstrates that cyclin B1 expression in ectopic endometrial cells significantly surpasses that in cells of the physiological lining. Immunohistochemical staining has pinpointed specific locations of excessive cyclin B1 accumulation within the cell nucleus, correlating with the heightened proliferation of endometriosis cells [66].

Comparative studies involving cervical epithelium from women with and without moderate or severe endometriosis reveal elevated expression of genes associated with endometriosis, including cyclin B1 (CCNB1) and cyclin G1 (CCNG1) [67]. Evidence supporting cyclin B1 overexpression in endometriosis includes elevated levels of specific substances. Research suggests these substances induce cyclin B1 overexpression, establishing a crucial cause-and-effect sequence in endometriosis pathogenesis. Compounds, including easily detectable esters, are implicated, enabling straightforward analysis in urine samples for scientific investigation [68]. In particular, this phenomenon is pronounced in the case of mono-n-butyl phthalate (MnBP). Patients diagnosed with endometriosis display higher urinary levels of MnBP compared to individuals in the control group. Furthermore, in vitro studies have shown that MnBP induces changes in the gene expression patterns of specific genes, such as BIRC5, BUB1B, and CDC20. These genes are closely linked to the regulation of cyclin B1 levels, underscoring the potential connection between MnBP exposure and disturbances in critical cellular processes related to cell cycle regulation [69]. Cyclin B1 has been demonstrated to be linked to the activity of a protein that serves as a substrate for ubiquitous nuclear casein and Cyclin-Dependent Kinase 1 (NUCKS1) [70]. NUCKS1 exhibits elevated expression levels in ectopic endometrium-like tissues. In vitro studies have revealed that inhibiting NUCKS1 significantly impairs cell proliferation and induces apoptosis in these tissues [71]. Considering the complexity of cyclin B1’s role in the proliferation of endometrial cells, it is essential to recognize that increased expression in diseased tissue does not necessarily imply it is the primary factor driving cell proliferation in the specific disease under investigation. Studies employing protein–protein interaction networks and analyzing differentially expressed genes have identified key genes within the networks of both normal endometrial lining tissue and ectopic endometrium-like tissue. In normal tissue samples, genes associated with the expression of cyclin B1 (encoded by CCNB1) were observed, along with CDK1 and BUB1. In contrast, in endometriosis tissue, genes such as CDC20, CCNB1, and CCNB2 played a central role. This finding suggests a significant alteration in molecular interactions in endometriosis. Therefore, analyzing cyclin B1 in isolation from other factors might be insufficient to fully understand its role in the context of endometrial cell proliferation [72]. This observation emphasizes the crucial role of the gene encoding cyclin B1 in the protein network of both ectopic endometrium and the physiological lining. In pathological conditions, the gene encoding cyclin B2 becomes significant, whereas its expression in the non-ectopic lining lacks central importance. This suggests that in the pathomechanism of endometriosis, cyclin B1’s function is complemented by the overexpression of cyclin B2 [66,73].

Furthermore, the expression of cyclin D1, which forms complexes with CDK4 and CDK6, is diminished in the presence of the overexpression of Polyadenylation Element Binding Protein 3 (CPEB3). In vitro studies conducted on primary human endometrial stromal cells demonstrated that elevated levels of CPEB3 resulted in decreased cell viability and inhibited cell cycle entry [74].

A crucial aspect underscoring the significance of cyclins in endometriosis is the role of specific regulators in influencing cyclin activity, particularly in initiating the cell’s transition to subsequent phases of the cycle. Even when cellular cyclin levels align with the specific physiological state of the uterus, the abundance of cyclin–CDK complexes, such as D-CDK4/6 and cyclin A-CDK2, may be elevated. This elevation depends on the presence of proteins (and the expression of the genes encoding them) that either degrade or inhibit the mentioned complexes or stimulate their formation. Notably, CDKIs targeting the CIP/KIP family, especially Cyclin-Dependent Kinase 1 (p21/WAF1) and p27kip1, play a significant role in this context [75,76]. The existing evidence implicates the p21/WAF1 inhibitor in endometriosis, albeit indirectly. This protein plays a role in regulating the cell cycle and is associated with p53, which halts the cell cycle, especially during the G1 to S phase transition in cases of DNA damage. Current reports suggest that a reduction in p21 levels likely leads to the growth and invasiveness of human endometrial stromal cells [75].

Moreover, p27kip1 (Cyclin-Dependent Kinase 1B inhibitor, CDKN1B) could also be significant within the framework of endometriosis. This inhibitor hinders cellular transition from G1 to S phase in healthy epithelial tissues [77]. Its levels cyclically fluctuate in healthy endometrial cells, being low during proliferation and rising in the secretory phase [78]. Ectopic endometriotic lesions exhibit remarkably low p27kip1 expression, suggesting progesterone as a potential therapeutic agent. Notably, estrogen exposure is a significant risk factor for endometrial cancer [76].

A definitive validation of the significance of cyclins in regulating ectopic endometrial proliferation lies in the demonstrated efficacy of inhibitors targeting specific cyclin–CDK complexes. In vitro experiments have unequivocally shown that these inhibitors effectively suppress cell division in tissue cultures, underscoring the pivotal role of cyclins in this process [79].

#### Angiogenesis in the Context of Cell Cycle Proteins in Endometriosis

The progression and sustenance of endometriotic lesions may necessitate the establishment of new blood vessels, a process known as angiogenesis [80]. Certain cyclins, particularly those pivotal in directing cell cycle advancement and proliferation, might additionally contribute to angiogenesis, thereby promoting the expansion of endometriotic lesions [81]. Therefore, it is postulated that while cyclins themselves may not be considered as a primary etiological factor in the development of endometriosis, deviations in their expression and functionality could potentially correlate with the aberrant proliferation and enhanced survival of endometrial cells situated outside the uterine environment [82].

There is substantiated evidence indicating that cyclins contribute to the mechanisms underlying angiogenesis in ectopic endometrium-like tissues. Specifically, the significance of the p21/WAF1 protein extends beyond its influence on endometrial cells; it is also linked to the angiogenic potential of tumors. This protein has the capacity to stimulate the expression of kinases responsible for the formation and functionality of endothelial cells within blood vessels [75,82]. Cyclins participate in the blood supply to these lesions by influencing the expression of Vascular Endothelial Growth Factor (VEGF), a typical angiogenic factor recognized for its role in various physiological and pathological connective tissues. The regulation of VEGF expression, influenced by NUCKS1 inhibition, is pivotal in promoting angiogenesis within endometriotic tissue and is essential for the development of endometriosis. In a broader context, NUCKS1, a gene associated with cell growth and proliferation, potentially plays a significant role in the advancement of endometriosis (Figure 3). The interconnection between NUCKS1, cyclin expression, and the previously discussed relationship highlights its indirect involvement in the regulation of cell cycle processes. This intricate network implicates both cyclins and CDKs in these cellular mechanisms [71].

One study by Sahraei et al. highlights notable differences in gene expression patterns between mesenchymal stem cells derived from menstrual blood (MenSC), specifically those isolated from patients with endometriosis (E-MenSC), and those from healthy women (NE-MenSC). The emphasis is particularly placed on the role of cyclin D1 in determining these differences. VEGF, known for its potent angiogenic properties, exhibited a substantial increase in E-MenSCs, suggesting its potential role in promoting the formation of new blood vessels. Furthermore, E-MenSCs displayed higher expression levels of cyclin D1, MMP-2, and MMP-9—factors associated with cell migration and invasion—compared to NE-MenSCs. Despite the increased cell proliferation observed in E-MenSCs, this study identified a significantly lower BAX/BCL-2 ratio, signifying reduced apoptosis in E-MenSCs compared to NE-MenSCs. Therefore, angiogenesis appears to be significantly associated with the expression of cyclin D1. However, this cyclin seems to exert a comprehensive and relatively broad pathological impact on endometriosis [83]. Additionally, in vitro studies have demonstrated that arcfiaflavin A, acting as an inhibitor of cyclin D1-CDK4, not only inhibits the expression of VEGF but also hampers angiogenesis and induces apoptosis in endometrial cells simultaneously [84].

The aberrant proliferation of endometriotic cells, mediated by cyclins, is suggested to be a critical component contributing to the pathogenesis of endometriosis. Increasing research underscores the crucial function of CDKI p21 in orchestrating this complex interaction. P21 impacts not only the cell cycle but also programmed cell death and the angiogenic potential in endometriotic lesions. Its inhibitory activity on CDKs, such as CDK1 and CDK2, slows the transition from the G1 to S phase, thus regulating the aberrant cell proliferation associated with endometriosis [75].

The study by Kim et al. highlights the pivotal role of p21-activated kinase 1 (Pak1) in endometrial cells, particularly under sex steroid treatment and in the presence of endometriosis. As a well-characterized serine/threonine kinase, Pak1 integrates essential signaling pathways for cellular survival. The investigation focuses on Pak1 expression in Ishikawa cells and the eutopic endometrium under the influence of Estradiol (E2) and Medroxyprogesterone Acetate (MPA). Notably, progestin (MPA) during the secretory phase leads to Pak1 downregulation, suggesting progesterone’s regulatory role. Elevated Pak1 expression in endometriosis-afflicted eutopic endometrium implies its potential involvement in the disorder’s initiation and progression [85].

In a comprehensive study by Li et al., p16 emerges as a pivotal regulator within the context of Tubal Metaplasia (TM) in the female genital canal. Specifically, p16 forms a binding affinity with cyclin D1-CDK4/6 complexes, exerting regulatory impact over the cell cycle’s G1-S transition. Nonetheless, when p16INK4a is absent, CDK6 is unhindered in promoting proliferation and inducing angiogenesis at its full potential (Figure 3) [86].

A study by Horree et al. reveals that the expression of p16 within the domain of TM is characterized by a consistently detectable mosaic pattern, particularly pronounced in endometrial TM. This unusual p16 expression is significant, indicating a possible premalignant propensity. According to this study, the unusual mosaic expression pattern of p16 in TM may serve as an identifier designating areas of tubal differentiation in the female genital tract. Furthermore, changes in p16 expression correlate with alterations in other critical cell cycle regulators, including cyclin A, cyclin E, p21, and p27, suggesting that TM causes disturbances in cell cycle transitions [87].

In the fourth generation of CDKIs, exemplified by compounds such as ON-123300 and TG02, increased potency compared to the first generation is evident. Notably, these compounds demonstrate a broad spectrum of activity, extending beyond the primary target to encompass additional pathways, including angiogenesis. Currently, these compounds are receiving rigorous scrutiny in both preclinical and clinical research studies [88].

**Figure 3 cancers-16-00836-f003:**
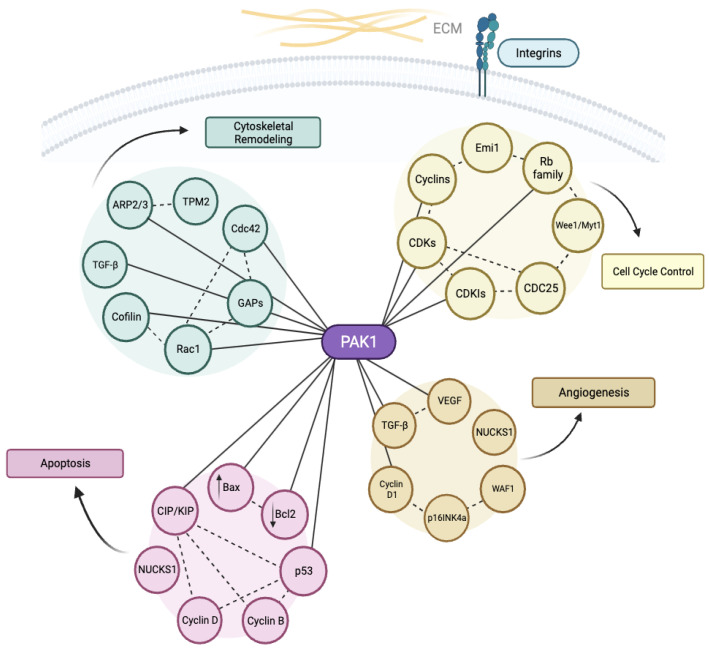
Illustration of the intricate protein–protein interactions within the molecular landscape of endometriosis. At the center of this network is the PAK1, which underscores its essential role in coordinating critical cellular processes. As evidenced, this protein exerts influence across a broad spectrum of functions, encompassing apoptosis, cytoskeletal remodeling, the control of the cell cycle, and angiogenesis through its interactions with the individual proteins illustrated in the diagram [89,90,91,92,93,94,95,96,97,98,99,100,101,102,103,104,105,106,107,108,109,110,111] (designed with BioRender: https://biorender.com/ (accessed on 27 November 2023)).

## 4. Cytoskeletal Proteins and Their Vital Biological Functions

The cytoskeleton is conventionally viewed as an intricate network of fibers and various protein structures. It plays a crucial role in providing spatial stabilization to the cell, maintaining its shape, orchestrating cytophysiological processes, and participating in intracellular transport, cell signaling, and cell division [112]. Additionally, the cytoskeleton plays a pivotal role in the EMT process [113], a meticulously controlled biological transformation through which epithelial cells undergo a shift to a mesenchymal state, marked by alterations in both cell structure and behavior. During EMT, the cytoskeleton undergoes dynamic reorganization, leading to alterations in cell adhesion and motility. This process is integral to various biological contexts, such as embryonic development and cancer metastasis, as it enables cells to migrate through tissues and invade surrounding structures. Understanding the cytoskeletal changes during EMT is essential for unraveling the mechanisms underlying pathological processes. Eukaryotes have the most complex cytoskeletons, consisting of three cytoskeletal polymers—actin filaments, intermediate filaments, and microtubules—acted upon by three families of motor proteins: myosin, kinesin, and dynein [113,114]. Detailed information about the characteristics and functions of key cytoskeletal proteins can be found in Table 2.

### 4.1. Cytoskeletal Proteins: Potential Contributors in Endometriosis

Cytoskeletal proteins, similar to cyclins, play a pivotal role in regulating and organizing cell division, which holds immense significance in the context of endometriosis, a condition characterized by the growth of endometrium-like tissue outside the uterus. The expression of specific cytoskeletal proteins is significantly elevated in both the ectopic endometrium and eutopic lining in women with endometriosis compared to healthy women. Furthermore, there are proteins with heightened expression levels that are specific to ectopic tissue [131]. For example, monomethyl auristatin E, a microtubule depolymerizing toxin, exhibits greater efficacy in eutopic cells compared to ectopic cells. This observation implies potential variations in the activity or function of cytoskeletal proteins or the pathways that they are engaged in between eutopic and ectopic endometrial stromal cells [131]. Similarly, comparing gene expression patterns in peritoneal endometriosis changes to the eutopic endometrium and peritoneum revealed that tissue remodeling genes associated with the cytoskeleton belonged to the most differentially expressed genes in peritoneal changes. Specific cytoskeletal-related genes, such as desmin and myosin heavy chain 11, significantly differed in terms of expression levels in peritoneal changes. Furthermore, the expression and localization of cytoskeletal proteins, such as desmin, alpha-actin, and h-caldesmon, in peritoneal changes appear to differentiate smooth muscle hyperplasia (increase in the number of muscle cells) from metaplasia (change in cell type) [132,133].

The similarity of endometriosis to cancer lies in the shared characteristics of unregulated cell proliferation and prolonged cell survival. In the cell cycle, actin and tubulin facilitate chromosome separation and cell division. During morphogenesis, they influence cell shape and polarity, and they strengthen stable cell–cell and cell–matrix connections by interacting with cadherins and integrins, respectively. Moreover, as cells migrate, they exert outward forces at the front while retracting forces at the rear, all of which are aspects of cell behavior frequently disrupted in cancer [134]. Therefore, it is reasonable to assume that the involvement of cytoskeletal proteins in endometriosis cells may bear some resemblance to these processes.

The importance of the cytoskeleton and its associated proteins becomes evident in the context of endometriosis cell migration, as they enable adhesion, ultimately influencing cell invasiveness. An illustrative case is the protein Talin-1, which plays a key role in facilitating cell–cell adhesion by bridging integrins to the actin cytoskeleton and activating integrins. Therefore, Talin-1 belongs to the category of cytoskeletal proteins actively involved in activating integrins and promoting interactions between the cell and the ECM [135]. The expression levels of both Talin-1 mRNA and protein are elevated in both ectopic endometrium and eutopic lining tissues in women with endometriosis in comparison to controls. The downregulation of this protein led to the inhibition of the cells’ ability to adhere, invade, and migrate, indicating the key role that Talin-1 plays in the progression of endometriosis and may play a role in the EMT process (Figure 4). Reducing the level of Talin-1 also leads to changes in the expression levels of other proteins, including *N*-cadherin, MMP-2, and integrin β3, while increasing the expression of *E*-cadherin [135].

Pyroptosis induced by Prostaglandin E2 (PGE2) has been shown to influence the progression of endometriosis by modifying the migration of pyroptotic cells and increasing the expression of High Mobility Group 1 Protein (HMGB1), the adhesion protein *E*-cadherin, and the cytoskeletal intermediate filament vimentin. These insights may pave the way for new therapeutic strategies using anti-inflammatory drugs to treat endometriosis [136]. *E*-cadherin and vimentin are undoubtedly strongly related to the cytophysiology of the cell cytoskeleton, as they are involved in the processes of cell adhesion and collective migration [137]. Their altered expression may, therefore, be related to cell movement and invasive features.

Retrospective studies have shown a relationship between neoplastic transformation and the role of cytoskeletal proteins in predicting the progression of endometriosis to ovarian cancer. The expression of the *CTNNB1* (β-catenin) gene may help to identify patients with high-risk endometriosis based on its significantly increased expression in a subgroup of patients compared to the control group [138]. β-catenin is associated with the cytoskeleton and involved in cell adhesion and the Wnt signaling pathway. In the Wnt/β-catenin signaling pathway, Multidrug Resistance Protein 4 (MRP4) influences Wnt/β-catenin signaling by stabilizing β-catenin, thereby affecting endometrial cells [139]. Furthermore, a robust positive correlation was observed between β-catenin and TGF-β2 protein levels in women with adenomyosis. The activation of β-catenin triggers the Epithelial–-Mesenchymal Transition (EMT) in endometrial epithelial cells via TGF-β2, and inhibiting TGF-β effectively suppresses the EMT-related effects of β-catenin activation [140].

Moreover, it is suggested that kaempferol plays a role in regulating genes associated with PI3K, particularly Phosphatase and Tensin Homologue (PTEN), and MMP9, thereby inhibiting endometrial cell implantation and ectopic formation [141]. Proteins linked to adhesion and implantation have also been detected in endometriotic cells through appropriate staining. For instance, the expression of ezrin and phosphorylated ezrin (Phospho-Ezrin) in endometriotic changes is notable. Endometriotic changes respond to immunostaining for the presence of these proteins (albeit to a varying degree in endocrine-active cells), while no differences in ezrin and phospho-ezrin expression have been observed in different types of endometriotic changes. However, an inverse relationship between severe dyschezia and ezrin expression intensity has been observed. Therefore, ezrin and phospho-ezrin may serve as markers for mechanisms related to the migration and attachment of endometriotic changes yet reveal little about their progression [142].

S100 proteins are also suspected of having a strong connection with endometriosis. They perform important functions in various biological processes through interactions with various cellular entities such as transcription factors, nucleic acids, enzymes, receptors, and the cytoskeletal system [143]. This group includes calcyclin, also known as S100A6. Calcyclin is a calcium-binding protein, important in the cytophysiology of the cytoskeleton. Therefore, it is believed that calcyclin may indirectly influence the dynamics of its proteins [144]. Another protein suspected of such associations is the Inverted Planar Cell Polarity Protein (INTU), which is highly expressed in endometrial tissue samples. It presumably has a role in cell orientation and polarization, which can be linked to the cytoskeleton and tissue organization (Figure 4) [145].

Increased levels of alpha-actinin 1 (α-actinin-1) binding proteins were observed in endometriosis cells, indicating heightened expression of this cytoskeletal protein in endometriosis. Conversely, the content of alpha- and beta-tubulin is diminished in patient groups with extragenital endometriosis compared to the control group. This implies that as ectopic endometrial foci relocate from their typical sites, the α-actinin-1 [9] content may rise, potentially signaling the increased migratory capacity of these cells. Additionally, a favorable prognosis in endometriosis could be attributed to the decrease in tubulin content, indicating a reduced rate of cell division [146].

#### 4.1.1. RNA Modulation of Cytoskeletal Dynamics in Endometriosis

On the other hand, long intergenic non-coding RNA 01133 (LINC01133) affects the cytoskeleton and morphology of endometriosis-related cells in such a way that increasing the expression of testis-specific protein kinase 1 (TESK1) corresponds to the phosphorylation and inactivation of actin-cutting cofilin. The inactivation of cofilin may lead to changes in the cytoskeleton, which, in turn, may affect cell migration. For this reason, cofilin is suspected of having a negative impact on cell migration [147]. In this context, it seems reasonable to say that cofilin, by cutting actin polymers, somehow “keeps in check” the cell susceptible to migration.

MicroRNA-142-3p (miR-142-3p), which is important in the regulation of the cytoskeleton, also seems to play an important role in the pathogenesis of endometriosis. After increasing the expression of miR-142-3p in vitro, several proteins involved in the regulation of the cytoskeleton fall into a state of reduced expression. This concerns ROCK2 (Rho-Related Protein Kinase 2), a protein involved in controlling cell shape, motility, and contraction. Furthermore, CFL2 (Cofilin 2); RAC1 (botulinum toxin C3 substrate related to Ras 1), which is involved in cell signaling and maintaining cell shape; and WASL (Neural Wiskott–Aldrich Syndrome Protein-Like), which plays a role in the reorganization of the cytoskeleton actin, are affected (Figure 4) [148]. Moreover, increasing the expression of miR-142-3p leads to significant functional changes in St-T1b cells. These include a reduction in the size of vinculin plaques (which are involved in cell adhesion to the ECM), reduced cell migration through fibronectin-coated filters, and the reduced ability of cells to contract in collagen I gels [148]. Consequently, miR-142-3p significantly modulates cytoskeletal dynamics and regulation, impacting cell behavior and potentially contributing to the invasive phenotype characteristic of endometriosis. This is particularly intriguing in the context of cancer research, where abnormal miRNAs like miR-142-3p have been found to influence various critical pathways in CRC, affecting processes such as cell proliferation, apoptosis, EMT, invasion, and metastasis [149].

It was also observed that overexpression of miR-145, a molecule known to be dysregulated in endometriosis, causes the downregulation of genes such as FASCIN-1, associated with cytoskeletal elements, pluripotency factors such as SOX2 and MSI2, leading to changes in cell behavior, including increased invasiveness in the Matrigel Invasion Assay. Therefore, the dysregulation of miR-145 expression plays a role in regulating the behavior of endometriosis cells by targeting genes related to cytoskeletal elements, pluripotency factors, and protease inhibitors (Figure 4) [150].

In the context of the role of cytoskeletal proteins, it should be added that the overexpression of miR-503 weakened the contractility of the ECM by suppressing the Rho/Rho-related helical protein kinase pathways that are involved in cell contractility. Meanwhile, miR-503 is epigenetically suppressed in endometriotic cyst stromal cells, which contributes to cell cycle deregulation in terms of cyclin D1 inhibition, apoptosis induction (Bcl-2 suppression), angiogenesis inhibition (VEGF-A suppression), and the attenuation of ECM contractility. Thus, miR-503 may play a role in the pathogenesis of endometriosis by influencing these cellular processes, including the regulation of cyclin D1, but also influences, among others, the cytoskeleton [84].

#### 4.1.2. Genes in Endometriosis: Cell Adhesion and Actin Cytoskeleton

It should be emphasized that among the genes showing increased expression in endometriosis, the vast majority are related to cell adhesion, extracellular exosomes, and actin binding. Notably, the ITGA7, ITGBL1, SORBS1, and IGHM genes have been identified as potential key factors in understanding the invasive nature of endometriosis, its recurrence, and potential therapeutic targets [151]. These genes shed light on the molecular mechanisms underlying this disease.

Genes with increased expression in endometriosis are significantly enriched in the pathway related to the regulation of the actin cytoskeleton. In this context, the FN1, EGF, EGFR, RAC1, and JUN genes can be mentioned. The dysregulation of genes related to the actin cytoskeleton may contribute to the cellular abnormalities observed in endometriosis [152]. The MYC transcription factor, which had a regulatory effect on most of these genes, also had an effect on RAC1, a factor involved in various processes, including actin polymerization [152].

#### 4.1.3. TGF-β and EMT Interplay in Endometriosis

The stability of the mesothelial barrier, crucial for safeguarding the peritoneal cavity against endometriotic lesions, is susceptible to disruption through the activation of the EMT. EMT induces changes in both cell morphology and behavior, affecting cytoskeletal proteins like actin, thus influencing the stability of the mesothelial barrier [153]. Specific genes associated with endometriosis are proposed to directly regulate the actin cytoskeleton, essential for coordinating the integrity of the interepithelial barrier. This transition involves epithelial cells undergoing transformation, relinquishing their characteristic features and acquiring those typical of mesenchymal cells, including reduced polarity, intercellular contact, and heightened motility. The cytoskeleton, necessary for maintaining cell shape, organization, and mechanical properties, plays a crucial role [7]. The changes in cytoskeletal organization can potentially enhance the mobility and invasive traits observed in cells undergoing EMT, contributing to the development of endometriotic lesions. Unfortunately, it is not certain exactly which type of EMT occurs in endometriosis, but several different types of EMT have been suggested to potentially be involved in the disease. Hypoxia and estrogen are stimulatory signals that can activate the EMT process in endometriosis through different pathways. Many of these pathways involve cellular factors such as TGF-beta and Wnt. TGF-β, particularly TGF-β1, emerges as a crucial factor in endometriosis pathogenesis. TGF-β is known to induce the EMT process and influence the reorganization of the cytoskeleton, further highlighting its significance in the context of endometriosis [154]. It is known that both of these signaling pathways play a role in the reorganization of the cytoskeleton. Moreover, in the early stages of endometriosis development outside the uterus, the attachment of endometrium-like tissue fragments to the pelvic mesothelium is essential. TGF-β1 and adhesion molecules play a crucial role in the adhesion of endometrium-like tissue fragments to the mesothelium outside the uterus by modulating the integrins αV, α6, β1, and β4 through the activation of the TGF-β1/TGF-βRI/Smad2 signaling pathway [155]. TGF-β is also implicated in fibrosis and inflammation, which are commonly observed in endometriotic lesions [156]. In a study by Zubrzycka et al., which was conducted on women with endometriosis and healthy controls, researchers evaluated the expression of markers associated with the EMT process, which included TGF-β1, SMAD3, ILK, and miR-21, in endometrium-like tissues. The results from this study revealed alterations in the expression of these genes in both the eutopic and ectopic endometrium-like tissues of patients with endometriosis, along with higher levels of miR-21 in endometrial implants. Furthermore, negative correlations were observed between miR-21 expression and the examined genes, particularly those related to TGF-β1. These findings suggest a significant role for the TGF-β1-SMAD3-ILK signaling pathway, possibly associated with EMT, in the pathogenesis of endometriosis, and highlight miR-21 as a potential inhibitor of this pathway [157]. Additionally, a specific microRNA associated with EMT, miR-21, has been identified in TGF-β-induced EMT in human keratinocytes. This serves as a model for understanding the plasticity of epithelial cells in response to epidermal injury and skin carcinogenesis. MiR-21, known for its abundant expression, plays a significant role in carcinogenesis by targeting two tumor suppressors: TPM1 and programmed cell death-4 (PDCD4). This targeting influences various cellular processes, such as cell proliferation, microfilament organization, and anchorage-independent growth. Notably, microRNAs exhibit an ability to target distinct functions within different signaling pathways, contributing to multiple key events associated with tumor progression. Consequently, the targeting of microRNAs emerges as a promising therapeutic approach for both cancer prevention and treatment, with the potential to have a comprehensive impact on various aspects of tumorigenesis [158,159,160]. A summary of key cytoskeleton-associated factors in endometriosis and related mechanisms is presented in Table 3.

Summarizing the context of therapeutic applications, this section provides a concise overview of the emerging field of modulating cytoskeletal proteins for the treatment of endometriosis. The challenging condition, characterized by the abnormal growth of uterine tissue, drives the exploration of innovative therapeutic avenues. Notably, the microtubule-depolymerizing toxin monomethyl auristatin E displays varying efficacies in eutopic and ectopic cells, indicating distinct cytoskeletal dynamics. The heightened expression of genes related to cell adhesion, extracellular exosomes, and actin binding in endometriosis reveals promising therapeutic targets [131]. Key influencers, like Talin-1, impact endometriosis cell migration, with its downregulation significantly impeding cell adhesion, invasion, and migration. The dysregulation of genes associated with the actin cytoskeleton further contributes to cellular abnormalities in endometriosis [135]. MicroRNAs, including miR-142-3p and miR-145, add complexity to cytoskeletal dynamics. Existing data suggest that elevated miR-145 levels may therapeutically inhibit cell proliferation, invasiveness, and stem cell properties, positioning them as a key target for endometriosis. These effects are anticipated to impede ectopic lesion growth, making miR-145–mimicking reagents potential therapeutics. Positive outcomes from miR-145–centered therapies in animal models further support their crucial role in endometriotic cell behavior. However, clarifying miR-145 dysregulation in clinical subgroups is crucial, given the previous variability in results. A comprehensive investigation, considering diverse cell types and larger patient cohorts, is essential to accurately define miR-145’s role [148,149,150]. While promising, further research on larger populations and using animal models is necessary for a comprehensive understanding and potential translation into clinical applications. Delving into the interplay between cytoskeletal proteins, migration, and factors like Prostaglandin E2-induced pyroptosis forms the basis for innovative anti-inflammatory strategies in endometriosis treatment [2]. This understanding provides insights into potential interventions to regulate cell adhesion, migration, and proliferation, thereby potentially mitigating the invasive nature of endometriosis. The conclusion underscores the need for continued research and clinical trials to validate and explore the efficacy of these therapeutic strategies in the management of endometriosis.

## 5. Conclusions

This review provides a contemporary analysis of cyclins and cytoskeletal proteins and their roles in endometriosis. It highlights the potential therapeutic use of CDKIs, particularly fourth-generation variants with anti-angiogenic properties. The review emphasizes the overexpression of cyclins in endometrial lesions, suggesting their active participation in disease development. An intriguing area for exploration is the role of cyclins in EMT, resembling neoplastic processes. The synthesis of existing research deepens our understanding of endometriosis at the molecular level, focusing on cellular processes linked to EMT dynamics. The review underscores the significance of maintaining endometriotic changes through the equilibrium of the cellular cytoskeletal framework, governed by diverse molecules, including those with cytokine-like properties. TGF-β’s intricate involvement in endometriosis and ongoing investigations into micro RNA molecules hold promise for tailored therapeutic interventions. These insights contribute significantly to the evolution of endometriosis research and potential therapeutic strategies. 

## Figures and Tables

**Figure 1 cancers-16-00836-f001:**
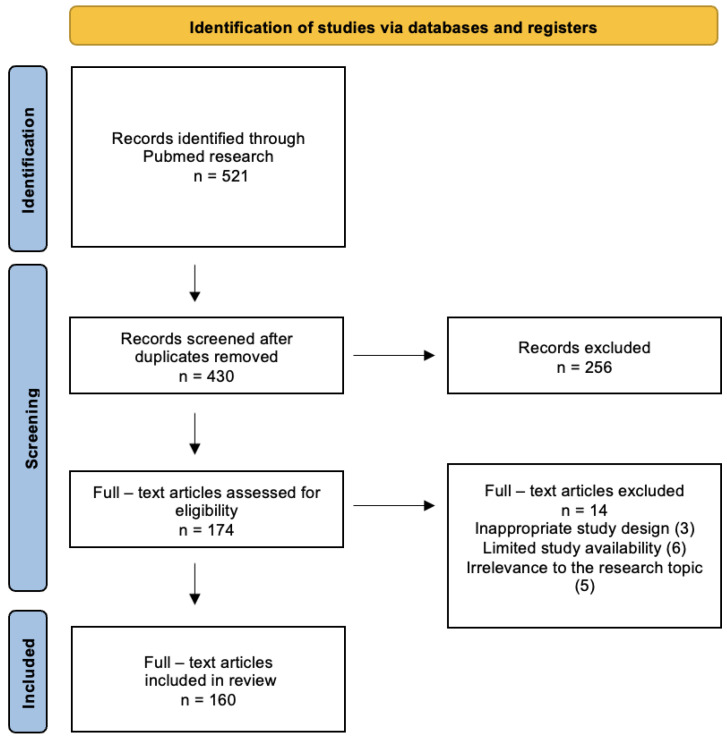
PRISMA flow diagram of this study. We identified 521 records through database searching. Following deduplication, 430 records were screened, from which 160 relevant studies were found and included in this review.

**Figure 4 cancers-16-00836-f004:**
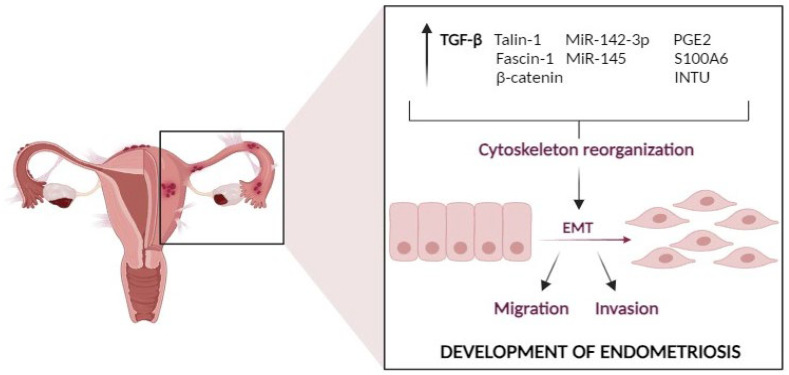
This figure illustrates the pivotal role of identified proteins and molecules, including TGF-β, Talin-1, Fascin-1, S100A6, PGE2, INTU, MicroRNA-142-3p, and MiR-145, in the reorganization of the cellular cytoskeleton, leading to the phenomenon of Epithelial–Mesenchymal Transition (EMT). EMT is characterized by the transformation of epithelial cells into a more mesenchymal-like phenotype, and the figure schematically presents the collection of identified proteins and molecules. These components are known to influence the cytoskeleton, subsequently triggering the process of EMT, which results in increased cell migration and invasion, playing a pivotal role in the pathogenesis of endometriosis. TGF-β (Transforming Growth Factor-Beta) is known to induce EMT and influence the reorganization of the cytoskeleton, ultimately contributing to the invasive nature of endometriosis. Talin-1 is a key player in cell adhesion and cytoskeletal reorganization, linking integrins to the actin cytoskeleton. Its increased expression affects the ability of cells to adhere and migrate, which is significant in endometriosis progression. Fascin-1 is involved in the organization of the actin cytoskeleton and influences cell morphology and behavior. It plays a role in cell migration and its altered expression can impact the invasive characteristics of endometriosis cells. S100A6, a calcium-binding protein, may indirectly influence the dynamics of cytoskeletal proteins and various cellular processes related to endometriosis. PGE2 (Prostaglandin E2)-induced pyroptosis modifies cell migration and the expression of High Mobility Group 1 Protein (HMGB1), *E*-cadherin, and the cytoskeletal intermediate filament vimentin, potentially contributing to endometriosis progression. INTU (Inverted Planar Cell Polarity Protein) is associated with cell orientation and polarization, which can be linked to the cytoskeleton and tissue organization. It may influence the directionality of cell movement. MicroRNA-142-3p significantly alters cytoskeletal dynamics and cellular behavior, potentially contributing to the invasive phenotype characteristic of endometriosis. MiR-145 overexpression leads to changes in cell behavior by targeting genes related to cytoskeletal elements, pluripotency factors, and protease inhibitors, ultimately affecting the invasiveness of endometriosis cells (designed with BioRender: https://biorender.com/ (accessed on 6 November 2023)).

**Table 2 cancers-16-00836-t002:** Cytoskeletal proteins and their vital biological functions.

Cytoskeletal Protein	Characteristics and Biological Functions	References
Actin	- Major component of cytoskeleton and microfilaments—switches between monomeric (G-actin) and filamentous (F-actin) states—facilitates cell division and cellular movement—maintains cell morphology—plays a significant role in Epithelial–Mesenchymal Transition (EMT)	Dominguez et al., Shankar et al. [115,116]
Profilin	- Binds G-actin and interacts with multi-acting binding proteins—associates with membrane lipids and cytoskeletal components—mediates interaction of actin microfilaments and microtubules	Pinto-Costa et al. [117]
Cofilin	- Regulator of actin dynamics and function—influences helical rotation of F-actin filament—regulates transportation and functionality of actin within the nucleus	Bamburg et al., Narita et al. [118,119]
Gelsolin	- Cytoplasmic regulator of actin organization—cuts off actin filaments and covers their ends—has various isoforms with potential roles in gene expression and serving as a plasma marker	Silacci et al., Bucki et al. [120,121]
Tropomyosin	- Regulates interaction between actin and myosin—crucial for muscle contraction and cytoskeletal dynamics in non-muscle cells	Gunning et al., Manstein et al. [122,123]
Microtubules	- Composed of alpha-tubulin and beta-tubulin heterodimers—utilize GTP hydrolysis for dynamic instability and polymerization—crucial for flexible spatial arrangements	Goodson et al. [124]
Myosin	- Motor protein that builds the sarcomere in muscle tissue—involved in generating force and cellular movement	Guhathakurta et al. [125]
Kinesins	- Affinity for microtubules—enable intracellular transport—create molecular motors for directional transport of cargoes	Hirokawa et al. [126]
Dynein	- Builds the kinetochore and participates in cell division—main variety is dynein 1, directed towards the minus side of microtubules	Gassmann et al. [127]
Other Cytoskeletal Proteins	- Keratin (mainly in epithelia)—vimentin (present in mesenchymal cells)—desmin (in muscle cells and mitochondria)—Nuclear lamina—junctional proteins (spectrin, ankyrin, dystrophin, septins)	Capetanaki et al., Zatloukal et al., Ridge et al. [128,129,130]

**Table 3 cancers-16-00836-t003:** Cytoskeleton-associated factors in endometriosis and related mechanisms.

Molecule/Protein Name	Experimental Model	Function	Related Genes/Pathways	References
Monomethyl auristatin E	Primary Human Endometriotic Stromal Cells (ESCs) obtained from patients with endometriosis	- Microtubule depolymerizing toxin specific to ectopic endometrium	-	Lavogina et al. [131]
Talin-1	Primary Human Endometriotic Stromal Cells (ESCs) obtained from patients with endometriosis	- Facilitates cell–cell adhesion, activates integrins - Increased expression observed in both ectopic endometrium and normal lining	N-cadherin, MMP-2, integrin β3, E-cadherin	Tang et al. [135]
Long intergenic non-coding RNA 01133 (LINC01133)	Endometriotic Epithelial Cell line 12Z	- Promotes cell proliferation and suppresses cell migration and invasion	TESK1, Cofilin	Yotova et al. [147]
MicroRNA-142-3p (miR-142-3p)	St-T1b cell line	- Affects cell proliferation, apoptosis, EMT, invasion, and metastasis	ROCK2, CFL2, RAC1, WASL	Börschel et al. [148]
MicroRNA-145 (miR-145)	Primary Human Endometriotic Stromal Cells (ESCs) obtained from patients with endometriosis	- Dysregulated in endometriosis and enhances the invasiveness of cells	FASCIN-1, SOX2, MSI2	Adammek et al. [150]
Prostaglandin E2 (PGE2)	Eutopic Stromal Cell Line (hEM15A)	- Induces pyroptosis - Influences cell migration	HMGB1, E-cadherin, vimentin	Sacco et al. [2]
miR-503	Human Endometriotic Cyst Stromal Cells (ECSCs)	- Downregulated in ECSCs and contributes to cell proliferation and resistance against apoptosis	Rho/Rho-related helical protein kinase pathways, cyclin D1, Bcl-2, VEGF-A	Hirakawa et al. [84]
TGF-β1	Ectopic and eutopic endometrium obtained from patients with endometriosis	- Correlates with the loss of the endometrial epithelial phenotype	SMAD3, ILK, miR-21, Integrins αV, α6, β1, β4, TGF-βRI	Zubrzycka et al. [157]

## Data Availability

The data presented in this study are available in this article.

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
