# Peer review of "Role of Cyclins and Cytoskeletal Proteins in Endometriosis: Insights into Pathophysiology"

_cancers, 2024, doi:10.3390/cancers16040836_

Round 1

Reviewer 1 Report

Comments and Suggestions for Authors

I appreciate the opportunity to review the manuscript entitled “Role of Cyclins and Cytoskeletal Proteins in Endometriosis: Insights into Pathophysiology” submitted to Cancers.

The authors conducted comprehensive analysis and reviewed the role of the cyclins and cytoskeletal proteins in endometriosis pathophysiology, both in relation to its onset and progression.

Reviewer Comments:

1.     Please prepare a list of abbreviations used in the manuscript.

2.     The authors state “The aim of this review is to highlight the significance of cyclins and cytoskeletal proteins in the context of the mentioned pathophysiological processes. The review focuses on the properties of cellular processes and their impact on the entire ectopic tissue during the course of endometriosis.” Nevertheless, the Materials and Methods section is entirely missing. Please, explain how the literature search was conducted. Which criteria were used for the selection of the publications included in the manuscript? What were the key words that authors used? Was there any limit in relation to the year of publication? Please make the appropriate flow-chart which describes the process of the publication search and selection for this review.

3.     Lines 20-26: Please, rephrase.

4.     Line 45: I would suggest the authors rephrase “body cavities”.

5.     Line 53: please omit the term “issues”.

6.     Lines 55-61: I would suggest explaining chronic pelvic pain in more detail, rephrase “life plans”. I am reluctant to accept that endometriosis impairs HRQoL in extreme cases, as it frequently diminishes HRQoL and jeopardizes patients’ life in its extreme cases (i.e., intestinal obstruction).

7.     Lines 74-76: Please, rephrase the term “non-uterine regions”.

8.     Lines 79-82: This classification is outdated.

9.     Lines 136-145: Please, provide appropriate references.

10.  The Introduction section is too long. Please make it shorter and much more concise, in relation to the topic of the study.

11.  The paragraph entitled “Primary Biological Functions and Regulatory Mechanisms of Cyclins in Cell Cycle Control” also must be more concise. I would suggest the authors summarize the data in the Table with corresponding references in it.

12.  The paragraph entitled “The Role of Cyclins and Cell Cycle Proteins in Evading Apoptosis, Driving Cell Proliferation and EMT in the Context of Endometriosis” should address role of the cyclins and cell cycle proteins in relation to endometriosis, as outlined in the subheading.

13.  Line 290: Please, delete the year of publication “2017”. The same stands for line 303, etc. throughout the manuscript.

14.  Line 475: there is an extra dot in this line. The same stands for line 487.

15.  Lines 569-589: Are all these data related to the reference #106? Please, summarize.

16.  The paragraph entitled “Cytoskeletal Proteins and Their Vital Biological Functions” needs to be more concise. I would suggest the authors summarize the data in the Table with corresponding references in it.

17.  Line 757: It seems that the end of the sentence is missing, or it is just a part of the sentence in line 759. Please, check it.

18.  Lines 782-783: Please, rephrase for clarity.

19.  Line 817: citations for “some studies” are missing in the text.

20.  Lines 826-827: Please, provide references for “previous research”.

21.  Lines 862-863: Citing reference for “in vitro studies” is missing.

22.  Lines 864-866: References for “retrospective studies” are missing.

23.  Despite the enormous number of cited publications, appropriate references are missing to support numerous claims throughout the text. Please, revise this.

24.  Please, omit the repetition of the “some studies” phrase, as well as “various” which repeats in the same sentence (Lines 899-891).

25.  Please, mention in a few sentences the role of cyclins and cytoskeletal proteins in other benign gynecological diseases, not only in endometriosis, if available.

26.  Please add the paragraph about potential clinical usage of the modulators of the cell cytoskeletal proteins in the therapy of endometriosis.

27.  Please prepare tables with the main findings of the included manuscripts in the review. Try to separate the studies on the animal and human models.

28.  Although the suggested length for review article is ≥4000 words, I would strongly recommend the authors to revise the entire manuscript and focus primarily on the subject-endometriosis. Thus, they will be able to eliminate too many text in certain sections of the manuscript and enormous number of references. This will make the entire text more interesting to the readers.

Based on the abovementioned comments this submission requires a major revision to meet the criteria required for publishing in Cancers.

Comments on the Quality of English Language

Minor editing of English language required.

Author Response

Thank you very much for taking the time to review this manuscript. Please find the detailed responses below.

Reviewer 2 Report

Comments and Suggestions for Authors

The manuscript  ‘Role of Cyclins and Cytoskeletal Proteins in Endometriosis: In-2 sights into Pathophysiology’  was reviewed with interest. This is a nice, refreshing manuscripts deserving publication.

This being said, and after considering the manuscript ‘Mikhaleva LM, Radzinsky VE, Orazov MR, Khovanskaya TN, Sorokina AV, Mikhalev SA, et al. Current Knowledge on Endometriosis Etiology: A Systematic Review of Literature.' the authors are strongly suggested to revise the manuscript as follows

-        A review of cyclins, EMT, and cytoskeletal proteins (i.e. as is done but without endometriosis)

-        Followed by a review on how this fits in endometriosis, especially if endometriosis is considered a benign tumor after some irreversible genetic or epigenetic incidents, eventually to consider with EMT.

-        Discuss the similarity with oncology.

Minor comments while reading

-        It is suggested to use endometrium-like tissue rather than endometrial tissue.

-        L46-49: very confusing terminology/ please keep endometriosis or endometriotic tissue and endometrium

-        L52-61: this manuscript is about pathophysiology, not about clinical symptoms

-        L62-78: Pathophysiology. Not clear why infertility is discussed since apparently unrelated to the pathophysiology of endometriosis. EMT: I would expect a comment on the metaplasia theory ( instead of implantation) and, eventually whether these (epigenetic) changes are permanent or reversible    

-        L86 ‘colloquially, these are referred to as superficial (psuperficial), ovarian (ovarian), and 86 deep (deep)’   this is the same as prevous sentence

-        L89-100 Only the implantation theory is considered, although incompatible with a series of observations such as clonality

Conclusions: the authors are congratulated for an innovative manuscript. Please consider the comments as suggestions to improve the readability and the citation index of the publication 

Comments on the Quality of English Language

Language is ok. 

Author Response

Thank you very much for taking the time to review this manuscript. Please find the detailed responses below

Reviewer 3 Report

Comments and Suggestions for Authors

In this review the authors describe the involvement of cyclins in the endometriosis etiology and development.

The review is well written, logically developing and touching the most important subjects to be learned for the better comprehension of the disease. In addition, the relevance of this matter as well as the interest for readers are undeniable.

I recommend the publication in the present form.

Author Response

(The authors gave the same response as above.)

Round 2

Reviewer 1 Report

Comments and Suggestions for Authors

I appreciate the opportunity to review again the manuscript entitled “Role of Cyclins and Cytoskeletal Proteins in Endometriosis: Insights into Pathophysiology” submitted to Cancers.

Although the authors made significant changes to the manuscript, I still have the impression that there is a lot of work to be done to make it interesting for the readers and important for the scientific community.

Some Specific Comments:

1.     Line 43: Please, rephrase “internal regions”.

2.     Although it has been changed, the entire Introduction section is still too long and contains unnecessary data. For example, is there any relationship between the type of endometriosis as described in lines 87-90 with role of cyclins and cytoskeletal proteins in endometriosis? If not, why is this classification mentioned in the Introduction?

3.     The Materials and Methods section has been inserted, but it is not properly written. Please, correct it.

4.     The suggestion for authors to provide Tables was to help condensate the text and reduce the overall manuscript volume. On the contrary, the text became more ambiguous with Table 1.

5.     Table 2. achieved suggested goal. Please, add into the column with the references the names of the first author.

6.     Cited publication should not be mentioned as “Y. Fang et al.”, but as following “Fang et al.”

7.     The Conclusions section should be written briefly.

8.     The total number of references is extremely big.

Based on the abovementioned comments this submission does not meet the criteria required for publishing in Cancers. I would strongly recommend the authors to significantly revise their work, condensate the main findings, reevaluate the literature and significantly modify the manuscript.

Round 3

Reviewer 1 Report

Comments and Suggestions for Authors

Lines 24-25 and lines 40-42: Definition of endometriosis is incorrect.

Lines 45-47: What is the true significance of these data for the study topic?

Lines 47-53: This text is confusing. Endometriosis is the term used both for ovarian endometriotic cysts and for peritoneal, bowel and bladder lesions.

It is nice that authors included Materials and Methods section in their work. Still, this section is too long and does not provide a brief explanation for the review genesis. Following sentence does not explain anything: “The literature was selected in order to establish causal relationships between variables.” The entire section must be condensed.

Lines 125-127: References explained as “from recent articles guided the inclusion of studies  spanning from the earliest publication in 1991 (Lees) to the most recent research in 2023 (Zhao et al.) “ are missing in the PRISMA diagram. The only citation authored by Lees was published in 1995 (reference 16) . Moreover, in this sentence instead of names of the authors in the brackets should be the reference numbers.

Lines 161-352: This part of the paper must be condensate significantly.

Lines 345-346: Table title should not be separated in two rows in such a manner.

Line 355: What is the meaning of “ectopic endometrium-like tissues in the context of endometriosis”?

Line 368: Is the first mention of the term EMT? If not, why the acronym is explained here?

Lines 413-416: Please, provide the references for “certain studies”.

Line 435: Reference is missing. Please insert it. Also, it would be nice to provide information which are “specific substances.”

Line 456: Please, check if “diseased tissue” is the right term.

Lines 457-462: Please, rephrase for clarity.

Lines 631-641: Where is the number for the corresponding reference Sharai et al? Please provide it.

Line 685: Please, rephrase “sex steroid treatment”. It is too ambiguous.

Lines 749-752: Are those data necessary for the readers to understand the topic?

Lines 756-757: Text is repeating “a condition characterized by the growth of endometrium-like tissue outside the uterus.” Moreover, this is not correct.

Lines 757-759: Please, rephrase for clarity.

Line 765: Using “eutopic” for peritoneum is wrong.

Line 769 and line 770: Please, explain “peritoneal changes”.

Lines 774-803: Although nice, Figure 4 has such a long explanation. It has an extremely long legend and does not have the simple necessary thing-a Title.

Line 821: The term “eutopic lining tissues” is unclear.

Line 962: In relation to claims about endometriosis development “outside the uterus” is there any difference between this and cervical endometriosis?

Lenes 1006-1008: Please, provide the reference.

Many of the claims in the manuscript are unclear or incorrect from the clinical point of view.

I still believe that a total of 160 citations is quite an enormous number for a single paper.

Based on the abovementioned comments, I would suggest the Editor to reject the manuscript and provide the authors with an option to rewrite, correct the entire manuscript, and resubmit it.